# The Short-Term Impact of Botulinum Neurotoxin-A on Muscle Morphology and Gait in Children with Spastic Cerebral Palsy

**DOI:** 10.3390/toxins14100676

**Published:** 2022-09-29

**Authors:** Nicky Peeters, Eirini Papageorgiou, Britta Hanssen, Nathalie De Beukelaer, Lauraine Staut, Marc Degelaen, Christine Van den Broeck, Patrick Calders, Hilde Feys, Anja Van Campenhout, Kaat Desloovere

**Affiliations:** 1Department of Rehabilitation Sciences, KU Leuven, 3001 Leuven, Belgium; 2Department of Rehabilitation Sciences, Ghent University, 9000 Ghent, Belgium; 3Inkendaal Rehabilitation Hospital, 1602 Vlezenbeek, Belgium; 4Rehabilitation Research Group, Vrije Universiteit Brussel, 1090 Brussels, Belgium; 5Department of Development and Regeneration, KU Leuven, 3000 Leuven, Belgium; 6Department of Orthopedic Surgery, University Hospitals Leuven, 3000 Leuven, Belgium; 7Clinical Motion Analysis Laboratory, University Hospitals Leuven, Pellenberg, 3212 Leuven, Belgium

**Keywords:** botulinum neurotoxin type-A, spastic cerebral palsy, gait, ultrasound, muscle morphology, treatment

## Abstract

Children with spastic cerebral palsy (SCP) are often treated with intramuscular Botulinum Neurotoxin type-A (BoNT-A). Recent studies demonstrated BoNT-A-induced muscle atrophy and variable effects on gait pathology. This group-matched controlled study in children with SCP compared changes in muscle morphology 8–10 weeks post-BoNT-A treatment (*n* = 25, median age 6.4 years, GMFCS level I/II/III (14/9/2)) to morphological changes of an untreated control group (*n* = 20, median age 7.6 years, GMFCS level I/II/III (14/5/1)). Additionally, the effects on gait and spasticity were assessed in all treated children and a subgroup (*n* = 14), respectively. BoNT-A treatment was applied following an established integrated approach. Gastrocnemius and semitendinosus volume and echogenicity intensity were assessed by 3D-freehand ultrasound, spasticity was quantified through electromyography during passive muscle stretches at different velocities. Ankle and knee kinematics were evaluated by 3D-gait analysis. Medial gastrocnemius (*p* = 0.018, −5.2%) and semitendinosus muscle volume (*p* = 0.030, −16.2%) reduced post-BoNT-A, but not in the untreated control group, while echogenicity intensity did not change. Spasticity reduced and ankle gait kinematics significantly improved, combined with limited effects on knee kinematics. This study demonstrated that BoNT-A reduces spasticity and partly improves pathological gait but reduces muscle volume 8–10 weeks post-injections. Close post-BoNT-A follow-up and well-considered treatment selection is advised before BoNT-A application in SCP.

## 1. Introduction

Spasticity occurs in approximately 60–80% of the children with cerebral palsy [1,2]. Therefore, spastic cerebral palsy (SCP) is the most common subtype. Spasticity is thought to be a result of imbalanced signals from the central nervous system resulting in impaired gross motor function, such as pathological gait, as well as in discomfort and pain [3,4]. Graham et al. described four stages of musculoskeletal pathology in SCP [5].

Botulinum Neurotoxin type-A (BoNT-A) is widely used during the first stage of musculoskeletal SCP pathology, which is characterized by hypertonia, as an effective treatment in reducing the hyperactive stretch reflex by temporarily blocking the release of Acetylcholine at the neuromuscular junction [6,7,8]. By reducing the hyperactive stretch reflex, BoNT-A aims to increase range of motion (ROM), reduce contracture development and improve function and gait [8,9,10,11]. Additionally, patient comfort, physical activity and social participation may improve [12]. Yet, the etiology of contractures is very complicated and not fully understood, and spasticity is definitely not the only contributing factor [13,14,15,16,17]. Still, BoNT-A injections remain the first line of treatment to prevent the child from developing to the second stage of contractures, which can only be surgically corrected [5].

As BoNT-A induces a local, temporary paralysis in the muscle, recent concerns have been raised regarding the impact of repeated BoNT-A injections on muscle morphology in young children with SCP [18,19,20]. A review by Multani et al. summarized evidence of muscle atrophy, fatty infiltration, loss of contractile tissue and increased connective tissue after BoNT-A treatment [18]. However, it should be noted that the highest levels of muscle atrophy and quality reduction with only partial recovery of muscle morphology and function at 12 months after BoNT-A injection have been reported in animal studies or studies including healthy volunteers [21,22,23]. Yet, knowledge on BoNT-A-impact on muscle morphology in a pediatric SCP population is important, specifically since muscle growth is already impaired in SCP from an early age on [24]. Moreover, multiple studies showed that muscle volume and muscle quality can be negatively influenced by BoNT-A [25,26,27,28,29,30]. It has also been shown that SCP muscles have a reduced number of satellite cells, suggesting that the regenerative capacity of the muscle might not be sufficient to ensure efficient muscle recovery, for example to overcome the potential paralysis-induced atrophy following BoNT-A injections [17,31,32].

Furthermore, early retrospective studies in young children with SCP who received one to five BoNT-A treatment sessions following an integrated approach (i.e., BoNT-A injections combined with stretching casts if relevant, intensive physical therapy, and orthotic management) showed beneficial effects on gait pathology at five to ten years of age (defined by three-dimensional gait analysis in 60 children with SCP) [33]. Moreover, BoNT-A treatment resulted in the reduced frequency of surgical interventions and postponed the need for surgery (defined in 424 children with SCP) [34]. The positive effects of BoNT-A, following this integrated approach, on gross motor function, including gait, have been confirmed by more recent studies [35,36]. A cohort study of 28 children with SCP showed that children who received repeated integrated BoNT-A treatment at a young age, improved or maintained their gross motor function at an older age [35]. However, these benefits have been recently questioned, in other studies that showed limited reductions in pathological gait [37,38,39]. BoNT-A-induced muscle weakness has been suggested as the potential cause for the lack of functional improvement, whereas SCP is already characterized by reduced muscle strength [19].

Despite the early reported positive effects of BoNT-A on gait and the potential to postpone more invasive treatments, the reported negative effects are concerning. Due to its temporary effect, BoNT-A injections have to be repeated. Different research centers [40,41,42], as well as previous BoNT-A treatment recommendations, reported repeated injections every six months or even less, while other centers reported intervals between repeated BoNT-A sessions of more than one year. Interestingly, the most recent recommendations advise an interval of at least one year [18,26,43]. Previous animal [22] and human [21,26] studies indeed showed that muscles are not recovered after six months. Multiple experts have asserted the need to revise treatment protocols as irreversible muscle atrophy may occur and the detrimental effects might become more severe after multiple injections [44]. Clearly, there is an urgent need for a critical reflection on the benefits and harms of BoNT-A in young children with SCP, which should be based on a comprehensive assessment of a multidimensional set of outcomes [18,23]. These outcomes should not only include clinical measures but also measures of the muscle morphology and objective measures of spasticity, as well as functional outcomes, such as gait analysis. A quantitative objective approach is important, as previous studies suggested that the benefits of BoNT-A can be overestimated when using subjective outcomes [18,45,46]. Even though the number of studies investigating the impact of BoNT-A in SCP has been rising in the past decade, previous research had some limitations. Small sample sizes, without control groups, were often included, subjective outcomes or isolated gait parameters were used and the focus was limited to only one muscle.

Therefore, the primary aim of the current group-matched controlled study was to investigate the impact of BoNT-A injections administered following an integrated therapeutic approach (combined with stretching casts if relevant, intensive physical therapy, and orthotic management) on the muscle morphology of two lower-limb muscles in children with SCP, aged between 3–11 years. The secondary aim was to confirm the impact of BoNT-A on spasticity and gait. The objective measures of the muscle morphology of the medial gastrocnemius (MG) and the distal compartment of the semitendinosus (ST) by three-dimensional freehand ultrasound (3DfUS), instrumented spasticity assessments and three-dimensional gait analysis (3DGA) were used to quantify the effects of BoNT-A. It was hypothesized that BoNT-A would reduce the muscle volume and negatively affect muscle quality. Furthermore, it was expected that multilevel integrated BoNT-A treatment would reduce the hyperactive stretch reflex and significantly improve gait kinematics of the ankle and knee in the sagittal plane.

## 2. Results

### 2.1. Participants

For the current study, the participants of two sub-studies on the effect of integrated BoNT-A treatment on muscle morphology and gait were pooled. This resulted in a total sample of 45 children (29 boys, 16 girls), with a median age of 7.1 years (5.7–9.1), who all had routine physical therapy and the use of orthoses as usual care. Twenty-five children, with an indication for BoNT-A injections in the ST and/or MG were included in the intervention group, while 20 children without an indication for BoNT-A were included in the untreated control group. The indication for BoNT-A treatment was set a by a multidisciplinary team based on clinical examination including assessments of ROM, spasticity, selectivity and strength, and the longitudinal evaluations of the pathological gait. Participants’ baseline characteristics and BoNT-A treatment details are summarized in Table 1.

All children in the intervention group underwent an assessment of muscle morphology by 3DfUS and received 3DGA to assess the level of gait pathology before and 8–10 weeks after the BoNT-A intervention. For one of the two sub-studies, the participants (*n* = 14, median age of 7.0 years (5.3–9.6), 6 boys and 8 girls, bilateral/unilateral 7/7, GMFCS level I/II/III 9/4/1 received an additional instrumented spasticity assessment at the same time points. The median interval between the baseline assessment (3DfUS + instrumented spasticity assessment) and integrated BoNT-A treatment was 0.2 (0.0–0.6) months. Due to practical reasons and the clinical follow-up, the pre-3DGA could not always be combined with the baseline assessments (3DfUS + instrumented spasticity assessment) required for this study. Therefore, the median interval between the pre-3DGA and the BoNT-A treatment was 1.1 (0.5–2.6) months. The median interval between the BoNT-A treatment and post-treatment evaluation (3DfUS + instrumented spasticity assessment + 3DGA) was 2.1 (1.9–2.4) months.

For the primary aim, 20 children with SCP, receiving no intervention and continuing usual care, were included in the control group. These children underwent 3DfUS assessments at baseline and after a period of 8–10 weeks, with a median interval of 2.7 (1.9–2.8) months but no 3DGA nor instrumented spasticity assessment.

The duration of the interval between BoNT-A treatment and the post-assessment in the intervention group and the interval between the baseline and post-assessment of the control group was not significantly different (*p* = 0.740).

As part of the integrated approach, 20 children received lower-leg casts immediately following the BoNT-A injections. These casts were applied for at least 10 days. Of the 20 children, 15 received additional removable upper-leg casts (as part of the integrated approach). Two children had full leg casts for at least ten days. Three children did not receive casting. 

At baseline, there was no difference in age (*p* = 0.144) between the intervention and control group. Body mass (*p* = 0.010) and height (*p* = 0.021) were significantly different. Therefore, the muscle morphology parameters were normalized to body dimension (body mass and height). 

### 2.2. Muscle Morphology

At baseline, no significant differences in normalized muscle volume (normalized to the product of body mass × height) or echogenicity intensity of the MG and ST were found between the intervention and control group.

The normalized muscle volume and echogenicity intensity before and after BoNT-A treatment are presented in Figure 1, Figure 2 and in the Appendix A. The normalized muscle volume of both the MG (*p* = 0.018) and the distal compartment of the ST (*p* = 0.030) showed a significant reduction of 5.2% and 16.2% (Appendix A) after BoNT-A treatment, respectively (Figure 1). No differences in the echogenicity intensity of the MG (*p* = 0.407) and ST (*p* = 0.145) were found within the intervention group (Figure 2).

No changes in muscle volume or echogenicity intensity were reported in the control group (Appendix A). The changes in MG (*p* = 0.118) and ST (*p* = 0.069) muscle volume post-BoNT-A were not significantly different from the evolution in muscle volume in the control group.

Absolute muscle growth rates are presented in Table 2. Muscle growth rates calculated over the period between the two assessments were not significantly different between the intervention and control group for both muscles (MG *p* = 0.346, ST *p* = 0.226).

All muscle morphology parameters, including differences’ scores and between-group comparisons are summarized in Appendix A.

### 2.3. Gait

The kinematic and spatial-temporal parameters are summarized in Table 3.

The ankle angle at initial contact (*p* = 0.002), the maximum dorsiflexion angle during both stance (*p* = 0.004) and swing phase (*p* < 0.001) significantly improved (while accounting for multiple testing, Bonferroni correction). The maximum dorsiflexion angle during swing exceeded the previously reported standard error of measurement (SEM) of 4.7 degrees (Appendix A) [47]. No significant improvements were found in knee kinematics in the sagittal plane following BoNT-A treatment. However, there was a tendency towards more knee flexion during stance (*p* = 0.065), whereas knee flexion at initial contact seemed slightly reduced (*p* = 0.048).

The gait profile score (GPS) showed a significant decrease (*p* = 0.005) after BoNT-A treatment, indicating less gait pathology. The median improvement of 1.76 degrees exceeded the minimally clinically important difference (MCID) of 1.6 degrees [48]. Cadence, walking velocity and stride length remained unchanged. 

The statistical non-parametric mapping (SnPM) analyses of the full kinematic ankle and knee waveforms revealed significant improvements in ankle kinematics, located in two clusters. One large cluster (*p* < 0.001) reached from 19–100% of the gait cycle and exceeded the SEM (ankle dorsi-plantar flexion, 3.3 degrees) [49]. Another small cluster (*p* = 0.0165, 0–1.99%) at initial contact did not exceed 3% of the gait cycle and was therefore not considered clinically relevant [50]. The knee kinematics showed a small cluster at the end of swing from 92–96% of the gait cycle, exceeding 3% of the gait cycle and the SEM (knee flexion-extension, 2.4 degrees) [49].

An overlay of the pre- and post-treatment kinematic waveforms for the ankle and knee is presented in Figure 3.

### 2.4. Instrumented Spasticity Assessment

The instrumented spasticity assessment performed in 14 children of the intervention group showed a significant decrease in muscle activation of the MG (*p* = 0.019) and medial hamstrings (including the ST) (*p* = 0.041) during the high-velocity passive stretches. All results are summarized in Table 4.

The difference in EMG signal between the low- and high-velocity stretches, a secondary outcome, showed a borderline significant decrease after integrated BoNT-A treatment in the ST (*p* = 0.060), exceeding the SEM (Appendix A) [51]. 

### 2.5. Clinical Examination and Range of Motion

The results of the clinical examination are summarized in the Appendix A. Maximal dorsiflexion (*p* = 0.302) and popliteal angle (unilateral) (*p* = 0.564) remained unchanged after BoNT-A injections.

## 3. Discussion

### 3.1. Summary of Findings

The primary aim of this study was to investigate the effect of integrated BoNT-A treatment on muscle morphology of the MG and ST. The second aim was to provide additional evidence regarding the effects on spasticity and gait. To achieve the study aims, an integrated set of objective assessments was used. We hypothesized that BoNT-A injections would result in reduced spasticity and improved gait but would induce muscle atrophy. The current results revealed that integrated BoNT-A treatment successfully reduced the hyperactive stretch reflex but also reduced muscle volume of the MG and distal compartment of the ST 8–10 weeks after BoNT-A injections. No changes in muscle quality were observed. Ankle kinematics during gait improved significantly, whereas effects on knee kinematics were limited. The study hypotheses were thus partly confirmed.

### 3.2. Comparison with Literature

#### 3.2.1. The Impact of BoNT-A on Muscle Volume

The current data showed a significant reduction (median change) in the normalized muscle volume of both the MG and ST of approximately 5.2% and 16.2%, 8–10 weeks after BoNT-A injections, which were applied as part of an integrated approach, respectively. Muscle growth rates, calculated over the period between the two assessments, were not significantly different from the control group. Several studies have reported similar rates of muscle atrophy after BoNT-A injections [21,22,23,25,26,27,28,29,30,52].

Alexander et al. reported reductions in MG volume, as absolute value and normalized to bone length at four (5.9%), 13 (9.4%) and 25 (6.8%) weeks, following the first single-level BoNT-a treatment to the MG. The greatest reduction, 13 weeks post-BoNT-A (9.4%), was slightly higher compared to the current data (5.2% median change, 8–10 weeks post-BoNT-A). Besides the MG atrophy, the (untreated) soleus muscle became hypertrophic in the sample of Alexander et al., resulting in an unchanged total volume of the plantar flexors. They assumed that the hypertrophy potentially developed to maintain plantar flexor strength and function. The reduction in volume of the MG combined with the hypertrophy of the soleus was also described by Williams et al. after five weeks post-BoNT-A [29]. They found an average reduction of 4.5% in the gastrocnemius (medial and lateral head). Five of those children received injections in the medial hamstrings as well, resulting in a volume reduction of 5.9%. All children in their study had a history of previous injections and received a BoNT-A injection in the MG as part of a multi-level treatment.

A recent study by De Beukelaer et al. was the first to quantify the BoNT-A effect on muscle morphology in an intervention group compared to a control group of children who did not receive BoNT-A treatment. They showed a reduction in anatomical cross-sectional area six months after the first BoNT-A injection, which was not observed in the control group of children, who remained BoNT-A naive during this period [26]. They did not find any significant between group differences in muscle volume following BoNT-A.

The decrease in ST volume reported in the current study is higher compared to the MG volume reduction and higher in comparison to previous studies of the distal lower limb muscles. The level of atrophy in the ST is more in line with a previous study by Van Campenhout et al. showing approximately 20.5% (79.5% of the initial volume) reduction in volume of the psoas two months after BoNT-A treatment [30], suggesting that proximal lower limb muscles might be more affected by BoNT-A. However, these findings should be confirmed in future studies including both distal and proximal lower limb muscles.

The current findings are in contrast with the results reported by Barber et al. [40], who showed an increase in absolute MG volume, reflecting an unchanged MG growth rate 12 months after BoNT-A. Williams et al. combined BoNT-A with strength training and also showed a significant increase in muscle volume of the hamstrings and plantar flexors (next to other lower limb muscles) six months post treatment [53]. However, it should be noted that improvements in absolute muscle volume observed in the latter studies were most likely a result of natural growth.

The differences in findings between the previously mentioned studies in SCP can be explained by the differences in study population, sample size, used methods, duration of follow-up, treatment protocol and content of usual care.

Besides the immediate impact of BoNT-A on muscle volume as described in the current prospective study, there are indications in the literature that previous BoNT-A injections in children who are not BoNT-A naïve may influence muscle morphology. A retrospective study by Schless et al. compared muscle volumes between children naïve to BoNT-A and children with a history of at least three previous BoNT-A treatments (median of five previous interventions) and showed that the absolute muscle volume was roughly 29% smaller in the children with a history of previous BoNT-A injections [25]. This may also further explain differences between the different studies. Longitudinal studies are required to further evaluate the impact of BoNT-A on muscle morphology and monitor the muscle-specific recovery profiles.

It should also be noted that the described alterations in muscle volume of children with SCP are less extreme compared to animal studies and studies with healthy adults [21,22,52]. This is potentially due to the aggressive treatment protocols that were applied in those studies.

#### 3.2.2. The Impact of BoNT-A on Muscle Morphology—Echogenicity Intensity

In the current study, no changes in echogenicity intensity were found, suggesting that integrated BoNT-A treatment does not negatively impact muscle quality in the short-term period. This is in contrast with a previous retrospective study by our research group showing that the MG echogenicity intensity of children with a history of BoNT-A was significantly higher compared to children who remained BoNT-A naïve, pointing towards long-lasting alterations in the muscles as a result of BoNT-A [25]. A more recent study by De Beukelaer et al. showed a tendency towards higher echogenicity intensity values in children with a history of previous injections compared to BoNT-A naïve children at baseline [26]. The fact that no alterations in echogenicity intensity were found in the current study could be explained by the fact that only the short-term effects of a single BoNT-A treatment session or short follow-up period were investigated. The higher echogenicity intensity values, as described in previous studies, suggest a reduced muscle quality, which may be caused by increased connective tissue, loss of contractile elements, fibrosis and fatty infiltration [13,14,26,27]. Indeed, multiple animal studies reported fibrosis or collagen deposition, increased lipid accumulation and loss of contractile elements after BoNT-A injections [22,44,54,55] and echogenicity intensity has been linked to these factors, contributing to declined muscle quality and contracture development [56,57,58]. Moreover, a causal relation between muscle fibrosis and non-responsiveness to BoNT-A has been suggested [43,59]. Yet, since echogenicity intensity represents an indirect measure of muscle quality and cannot decompose the contribution of different components, such as increased fat infiltration and connective tissue, the earlier retrospective results in humans should be interpreted carefully. The validity of echogenicity intensity is still not fully understood. It has been related to muscle composition in several pathologies; however, validation studies in SCP are lacking [58,60,61].

#### 3.2.3. The Impact of BoNT-A on Muscle/Tendon Length, Range of Motion and Spasticity

The secondary morphological outcomes (Appendix A) showed an increase in muscle belly length (*p* = 0.009) and total muscle-tendon unit length (*p* = 0.011) of the MG after BoNT-A injections, while these length data did not change for the ST. However, the maximum ankle dorsiflexion angle defined by goniometry during the clinical examination did not change post-BoNT-A. A previous investigation showed a similar increase in MG muscle belly length combined with unchanged maximum ankle dorsiflexion already at two weeks post-BoNT-A injection [62]. Additionally, the resting ankle angle during the 3DfUS shifted towards increased dorsiflexion (median increase of 10 (12.5–0) degrees dorsiflexion, *p* = 0.003) after integrated BoNT-A treatment. This could explain the increase in muscle belly and total muscle-tendon unit length. Additionally, a previous observational study by Malaiya et al. did not find an association between MG muscle belly length and maximum dorsiflexion angle in a cohort of 16 children with unilateral SCP [63].

For the instrumented spasticity assessments, the average EMG signal while stretching the muscle at high-velocity was significantly reduced post-BoNT-A compared to the baseline condition. This resulted in a (tendency) towards a reduced difference in EMG between high and low stretch velocity (which exceeded the SEM). These results confirmed previous findings that BoNT-A injections in the MG and ST successfully reduces the hyperactive stretch reflex [62,64,65].

#### 3.2.4. The Impact of BoNT-A on Gait

In contrast to the studies by Hastings-Ison, the current study found significant improvements in the overall gait pathology defined by the GPS [37,38]. The SnPM analyses of the full kinematic waveforms showed one large relevant ankle cluster, as well as one small cluster in the knee during swing. These improvements were confirmed by beneficial changes in gait features of the ankle (significant for ankle at initial contact and maximal dorsiflexion in stance and swing) and knee (tendencies for knee angle at initial contact and knee ROM). These findings are in line with a previous study [66]. Hastings-Ison et al. did not find any alteration in maximum ankle dorsiflexion or knee kinematics in the total group. However, they showed improved ankle dorsiflexion and increased knee flexion during mid stance in bilaterally involved children but not in unilaterally involved participants [37]. The current data were not separately analyzed for children with uni- and bilateral involvement due to lack of power. We previously performed a systematic review on the BoNT-A effect on gait and compared the BoNT-A effect on gait features with the effect defined by statistical parametric mapping in a group of 51 children with SCP who received multilevel BoNT-A treatment [66]. The latter study showed significant improvements at the ankle as well as at the knee joint, and these beneficial effects were also more pronounced at the ankle compared to the knee joint. Overall, we conclude that BoNT-A improved the gait of the children with SCP at 8–10 weeks post-BoNT-A.

### 3.3. Frequency of BoNT-A Administration

As highlighted by the study of Multani and colleagues, the BoNT-A injection frequency is an important matter of debate [18,67]. Studies by Kanovsky and Hastings-Ison showed that BoNT-A treatment once per year was effective with fewer adverse events compared to multiple injections per year [41,42]. These findings were supported by animal studies showing more muscle weakness and atrophy, increased loss of contractile tissue and exacerbated fibrosis after more frequent injections [23,44,54]. As the muscle might need more time to recover, even if the clinical effect of tone reduction is dissolved, the duration between injections should be based on assessments of the muscle, preferably at both the microscopic and macroscopic level [23,44,52].

### 3.4. Combined Outcome Assessments and Clinical Implication

This is the first study that combined the assessments of muscle morphology and gait, in order to critically review the benefits and potential harm of integrated BoNT-A treatment. The observed clinical benefits included reduced spasticity of the plantar flexors and knee flexors and improved ankle motion during gait but had rather limited effects on knee kinematics. On the other hand, the muscle volume was reduced 8–10 weeks post-BoNT-A treatment. These pros and cons should be weighted before the selection of a (repeated) BoNT-A treatment for young children with SCP. The recovery profile of the individual patient should be objectively monitored. The current results revealed muscle atrophy in the short-term period following BoNT-A injections and other studies highlighted that muscle recovery is still ongoing six months post-BoNT-A [26,27,40]. These findings suggest that muscle atrophy lasts longer than the clinical effects, such as tone reduction [23,52]. An animal study by Minamoto et al. hypothesized that the effect and potential negative effects of BoNT-A are much stronger if injected into a muscle which has not completely recovered [44]. The current evidence implies that BoNT-A injections should be provided with an interval of at least 12 months. As stated earlier, the first BoNT-A injections seem to be most distinctive in both the beneficial functional effects and the harmful effects on the muscle. Therefore, muscle recovery is especially crucial after the first injection and an interval of at least 12 months before the second injection is advised.

### 3.5. Future Perspectives

The observed muscle atrophy, after approximately two months post-BoNT-A in the current study and six months post-BoNT-A in a previous study [26], represents the rather short-term effect of BoNT-A on muscle morphology. Hence, no conclusion can be drawn on the long-term effects. More research is needed to define the time course of muscle recovery. Future studies should investigate the long-term effects and include the microscopic and molecular analyses of intrinsic muscle properties and functional evaluations in order to critically appraise the benefit–harm balance of BoNT-A treatment. Thereby, more research on dosage dependent effects, the frequency of BoNT-A injections and treatment history is relevant. In order to gain more insights into the patient- and or even muscle-specific response to BoNT-A and in order to properly identify responders and non-responders, the objective measures of muscle spasticity as well as morphology and function, such as gait [37,39,64,68], should be combined in future studies and implemented in clinical practice. These findings may improve efficient treatment planning.

Most studies, including the current study, focused on the effects of BoNT-A in ambulatory children. Therefore, future studies should also include participants with less functional capacity in order to evaluate the benefits and potential harms in non-ambulatory children.

### 3.6. Limitations

Some limitations have to be acknowledged. This study included a non-randomized control group. As BoNT-A has been proven to be effective in reducing spasticity and time is running out for growing children with spasticity, as this may lead to impaired musculoskeletal development, it was considered unethical to temporarily withdraw treatment from patients. Therefore, randomization procedures were not possible. We minimalized bias by group-matching age and GMFCS level. The control group presented with slightly lower spasticity levels compared to the intervention group at baseline, which was not surprising since these children had no indication for BoNT-A treatment. No control data were available for gait and spasticity assessments. However, gait and spasticity parameters were evaluated using previously reported reliability values [47,48,51]. In the current study, BoNT-A was applied as part of an integrated approach, whereby BoNT-A injections were combined with casting, orthotic wear and intensive physical therapy. Hence, no conclusions can be drawn on the contribution of the isolated treatments. 

Due to the deep origin of the ST, only the distal compartment could be visualized by 3DfUS, even though both compartments were targeted by BoNT-A [69]. Muscles were only assessed in resting length, therefore no conclusion on alterations in extensibility can be drawn.

The assessments were performed by three different assessors. However, because all assessors were well-trained (i.e., minimum of two years of experience), regular comparisons of performance to ensure standardization were made and because the pre- and post-assessments were always performed by the same assessor, we do not believe that this could have influenced our results.

This study was ongoing during the onset of the COVID-19 pandemic. As the majority of participants was included after the first COVID-19 outbreak and the application of an additional inclusion criterion stating that the physical therapy had to be stable for at least three months, we believe that the pandemic did not influence the outcomes of the current investigation.

## 4. Conclusions

This study including a comprehensive set of objective measures showed that BoNT-A injections, when administered as part of an integrated approach (i.e., BoNT-A injections combined with stretching casts if relevant, intensive physical therapy and orthotic management), results in effective tone reduction, with positive effects on ankle kinematics but limited effects on knee kinematics during gait, combined with muscle atrophy, 8 to 10 weeks post-BoNT-A. There is a need for evidence-based guidelines rather than guidelines based on clinical experience in order to further evaluate the impact of BoNT-A. Close follow-ups with an integrated set of objective measures are required to gain more insight into the patient-specific response and recovery profile. Furthermore, patient-specific goals and expectations should be carefully evaluated and contemplated before BoNT-A treatment is planned in young children with SCP.

## 5. Materials and Methods

This study included participants of two different projects. Both projects were approved by the local ethics committee of the University Hospitals Leuven, Belgium (s59945 TAMTA project, s62187 and s62645 3D-MMAP project) (Ethische Commissie Onderzoek KU Leuven/UZ Leuven). Written informed consent was obtained from the parents or legal guardian. Participants were included between May 2019 and June 2022.

This trial is registered at ClinicalTrials.gov, identifier: NCT05126693.

### 5.1. Participants

Children aged between 3–11 years, GMFCS levels I-III with SCP, scheduled for BoNT-A in the MG and/or ST, as part of a multilevel treatment, were included. Participants were recruited via two parallel prospective studies. Both studies investigated the effect of integrated BoNT-A treatment on muscle morphology and gait, while one study also evaluated the effects on spasticity (s59945 TAMTA project). The selection criteria for the participants were similar in both studies. The exclusion criteria were: any signs of dystonia or ataxia in the lower limbs, severe muscle weakness or poor selectivity in the plantar flexors or knee flexors and cognitive problems that would impede the measurements, intrathecal baclofen pump, previous muscle surgery in the lower limbs or selective dorsal rhizotomy, previous bony corrections in the lower limb within two years before assessments, previous BoNT-A treatment within six months before the baseline assessment or within six months before the pre-BoNT-A 3DGA. Additionally, the physical therapy frequency had to be consistent for at least three months before inclusion. The participants were measured twice, before and 8–10 weeks after BoNT-A treatment to evaluate the short-term effects. All children (of both studies) received 3DfUS assessments and a 3DGA with the clinical examination of the lower limbs. A subset of children (i.e., the participants of one of the two studies) underwent an additional instrumented spasticity assessment in order to evaluate the BoNT-A-induced tone reduction. During the follow-up, all children received usual care, including intensive physical therapy, orthotic management and oral tone reduction if needed.

An age- and functional severity-matched (group-matching based on age and GMFCS level) control group, meeting the same inclusion criteria as the intervention group, but not receiving any intervention apart from their usual care, was included. The participants of the control group were recruited via Cerebral Palsy Reference Center of the University Hospitals Leuven, Belgium and the Inkendaal Rehabilitation Hospital Vlezenbeek, Belgium. All children eligible for inclusion were measured (*n* = 30, Figure 4); however, the final data selection was based on accurate matching with the participants in the intervention group. The participants in the control group underwent 3DfUS measurements only, with an interval of 8–10 weeks between the two assessments. Participant characteristics are summarized in Table 1, showing no structural differences, as assessed by a Mann–Whitney U test, in the maximum dorsiflexion angle, popliteal angle and spasticity levels assessed by the modified Ashworth scale.

An overview of the recruited and eventually included subjects and data availability is summarized in Figure 4.

### 5.2. Procedure

The most affected leg, based on the most recent clinical examination (highest modified Ashworth score for the ankle and knee joints), was assessed. The measures of body mass, height, total leg length and lower leg length were taken. A standardized anamnesis was performed to request treatment history and usual care details.

Muscle selection for BoNT-A injection was primarily based on the most recent clinical examination and multidisciplinary evaluation of the 3DGA data. BoNT-A, Onabotulinum toxin A (Botox^®^, Allergan, Diegem, Belgium), injections were applied by a pediatric orthopedic surgeon under general (mask) anesthesia, guided by ultrasonography or palpation [30,69]. A dose of 100 units of BoNT-A was diluted in 5 mL of saline. The BoNT-A dilution was injected at multiple sites, in areas with the largest concentration of motor endplates in the muscles selected for treatment [69]. A maximum of 50 units was injected per site.

BoNT-A is suggested to be more effective when part of an integrated approach, i.e., combined with casting, orthotic wear and physical therapy [68]. Therefore, lower leg casts and removable upper leg casts were applied as indicated. The majority of children used day and night orthoses (Table 1) and all children received intensive physical therapy in the 8–10 weeks following BoNT-A treatment.

All additional study assessments (3DfUS, clinical examination, instrumented spasticity assessment and 3DGA) were performed by three trained assessors. The pre- and post-assessments were always performed by the same assessor.

### 5.3. Assessments

#### 5.3.1. Muscle Morphology

The muscle morphology of the MG and the distal compartment of the ST muscle were assessed by 3DfUS [70]. This technique, combining B-mode ultrasonography with three-dimensional motion tracking, is proven to be valid and reliable in muscle morphology assessments in children with SCP in previous studies [71,72,73].

Ultrasound assessments were performed while the participant was placed in a prone position. The lower leg was supported by a triangular cushion to allow slight knee flexion (approximately 25 degrees) and avoid bi-articular stretch on the MG and ST. The ankle was in resting position without fixation. The joint angles were measured by goniometry.

Large amounts of acoustic transmission gel were used to assure smooth movement of the transducer and optimal contact between the transducer and skin. An additional curved gel pad, the Portico, was used to reduce muscle deformation during the acquisition [74]. Muscle images were recorded by Stradwin software (version 6.0; Mechanical Engineering, Cambridge University, Cambridge, UK) while moving the transducer from muscle origin to tendinous insertion. The acquisitions were repeated in case of bad quality images, movement or muscle contraction or when the bony landmarks or muscle borders were missed. 

After data collection, the muscle images were processed by a trained assessor using the Stradwin software (version 6.0; Mechanical Engineering, Cambridge University, Cambridge, UK). Muscle volume (in millimeters) was estimated by the cubic planimetry technique based on segmentations that were manually drawn along the inside of the muscle borders in 5–10% of the transverse 2D muscle images [75]. Additionally, echogenicity intensity, as a proxy of muscle quality, was based on the average of the interpolated reconstruction of the muscle and expressed as a gray value on an 8-bit scale (values between 0–255), whereby higher values (whiter images) indicated reduced muscle quality [56,58]. Muscle belly lengths were calculated as the Euclidean distance between the muscle origin (MG: posterior surface of medial femoral condyle, ST: ischial tuberosity) and the muscle-tendon junction. The tendon length of the MG was calculated between the muscle–tendon junction and tendinous insertion. The MTU length of the MG was calculated by adding the muscle belly to the tendon length.

In order to compare children with different body dimensions, muscle volume was normalized to the product of body mass × height and muscle and tendon lengths were expressed as a percentage of total leg length.

The absolute muscle-specific growth rates for muscle volume during the period between the two assessments were calculated by the following formulas [26,40]: $$\mathrm{growth}\;\mathrm{rate},\;\mathrm{muscle}\;\mathrm{volume}=\frac{\mathrm{muscle}\;\mathrm{volume}\;\left(\mathrm{mL}\right)\;\mathrm{POST}-\;\mathrm{muscle}\;\mathrm{volume}\;\left(\mathrm{mL}\right)\;\mathrm{PRE}\;}{\mathrm{age}\;\left(\mathrm{months}\right)\;\mathrm{POST}-\;\mathrm{age}\;\left(\mathrm{months}\right)\;\mathrm{PRE}}$$

The primary outcomes included the normalized muscle volume and echogenicity intensity of the MG and distal compartment of the ST and the absolute muscle volume growth rates. The secondary outcomes, muscle, tendon and muscle-tendon unit lengths and normalized muscle volume growth rates are included in the Appendix A.

#### 5.3.2. Gait Analysis

Gait, overground barefoot walking at a self-selected speed, was assessed by 3DGA in a 10 to 12 camera (Oxford Metrics, Oxford, UK) gait lab (Vicon Motion Systems, Oxford, UK). Twelve reflective markers were placed on anatomical landmarks according the Vicon Plug-In-Gait lower limb model [76]. Ground reaction forces were measured with force plates (Advanced Mechanical Technology Inc., Watertown, MA, USA) that were embedded in the walkway. During the 3DGA, kinematic data (joint angles for ankle, knee and hip and the segmental orientation of the pelvis and foot), kinetic data (joint moments and power) and the surface EMG data of the medial gastrocnemius, soleus, tibialis anterior, rectus femoris, vastus lateralis, the medial and lateral hamstrings and gluteus medius were collected for each leg. The kinematic data of the most affected leg (same as for the 3DfUS and instrumented spasticity assessments) and a selection of spatial-temporal parameters were included in this current investigation but the kinetic and EMG data were not included in any of the analyses.

Gait cycles were identified and manually marked using the Vicon Nexus software (Oxford Metrics, Oxford, UK). All trials were exported and imported to a custom-made Matlab script (The MathWorks, Natick, MA, USA, version 2020b). All gait cycles were judged on quality. Trials were excluded in case of artefacts, gaps in data or indications of inaccurate marker placement. Furthermore, the knee varus–valgus angle was evaluated and trials with absolute knee varus–valgus ROM exceeding 15 degrees, a knee valgus angle exceeding −10 degrees, or trials that were not representative of the participants’ gait pattern were excluded [50]. As many good-quality steps as possible were used to create average trials for the ankle and knee joint kinematics.

The GPS, and specific kinematic gait parameters that were found to be sensitive to detect BoNT-A-induced changes, were extracted per joint [66]. Additionally, the spatial-temporal parameters cadence, walking velocity and stride length were calculated. Finally, the full kinematic waveforms were exported for pre–post comparison.

The primary outcomes included the kinematic waveforms, BoNT-A-sensitive kinematic gait parameters [66], GPS and spatial-temporal parameters. The secondary outcomes of elaborate kinematic parameters are summarized in the Appendix A.

#### 5.3.3. Spasticity Assessment

A subset of participants (n = 14) underwent an instrumented spasticity assessment in order to objectively evaluate the change in the hyperactive stretch reflex during passive stretch [77]. This method has been previously described by Bar-On and colleagues [77] and is proven sensitive enough to detect treatment-induced changes [62,64,65].

Four low-velocity (over 5 seconds) passive stretches of the plantar flexors and medial hamstrings were performed over the full ROM. Next, four full ROM high-velocity passive stretches were performed, whereby the stretch was performed as fast as possible. The interval between the stretch repetitions was at least 7 seconds to avoid post-activation depression. During the stretches, muscle activation was recorded by surface EMG (Zerowire Cometa, Milan, Italy)

The raw EMG signals were filtered by a 6th order zero-phase Butterworth bandpass filter (20–500 Hz). The root mean square envelope of the EMG signal (RMS–EMG) was computed after application of a low-pass 30 Hz 6th order zero-phase Butterworth filter.

All data analyses were carried out with MATLAB Software (The MathWorks, Natick, MA, USA, version 2020b). The quality of the individual stretches were checked using custom-made software. Stretch repetitions were excluded in case of inconsistent stretch velocities, poor surface EMG data, out-of-plane movement or indications of insufficient participant relaxation (constant EMG activity of the antagonist).

In order to quantify the alteration in muscle tone after BoNT-A treatment, the average RMS–EMG expressed in millivolts was calculated as the square root of the area underneath the RMS–EMG time curve, divided by the duration of the time interval, during the high-velocity stretch. The interval started 200 milliseconds prior to the time point where maximum stretch velocity was reached and ended at the time corresponding to 90% of the full ROM. This value was referred to as the muscle activation during a high-velocity stretch. All BoNT-A-induced changes were compared to previous reported SEM or BoNT-A-induced values [51,62,64,65,77].

The average RMS–EMG expressed in millivolts during a high-velocity stretch was used as the primary outcome to assess alterations in muscle tone. Additionally, the increase during the high-velocity stretch was calculated by subtracting the muscle activation during a low-velocity stretch, which was calculated following the same method. This value was called the change in muscle activation and was included as a secondary outcome. This secondary outcome together with the RMS–EMG during a low-velocity stretch are enclosed in the Appendix A.

#### 5.3.4. Clinical Examination

A standardized clinical examination including assessments of ROM, spasticity, selectivity and strength was performed during the pre- and post-BoNT-A assessments of the intervention group and during the pre-assessment of the control group. These data were used to describe and compare both groups at baseline. The maximal dorsiflexion angle and popliteal angle (unilateral) were assessed by goniometry. The level of spasticity was assessed by the modified Ashworth scale and modified Tardieu scale [78,79]. Muscle strength was assessed by manual muscle testing and selectivity was assessed by the Selective Control Assessment of the Lower Extremity [80,81]. The evaluations of strength and selectivity were not included in the current investigation.

### 5.4. Statistical Analysis

The normality of data distribution was checked by a Kolmogorov–Smirnov test (*p* < 0.05). Since most parameters were not normally distributed, non-parametric statistical tests were performed.

Participant characteristics, maximal dorsiflexion angle (measured with the knee extended), popliteal angle (unilateral), and normalized muscle morphology parameters of the intervention and control group were compared at baseline by a Mann–Whitney U test (Table 1).

A Wilcoxon signed rank test was used to evaluate (within) treatment effects in muscle morphology, instrumented spasticity assessment outcomes and spatial-temporal parameters of the 3DGAs (alpha = 0.05). A Bonferroni correction was applied for the gait kinematic parameters per joint (ankle: alpha 0.05/4 = 0.0125, knee: alpha 0.05/3 = 0.0167). Gait parameters were considered meaningful if statistically significant changes were noted that were exceeding the SEM or MCID [47,48].

Additionally, median difference scores (x_post_−x_pre_) for muscle morphology parameters were calculated in both the intervention and control group and between-group differences were tested by a Mann–Whitney U test. Additionally, the changes were expressed as a percentage change of the baseline values for both groups, which are summarized in Appendix A.

Paired t-tests of the ankle and knee kinematics over the entire gait cycle using SnPM analysis were performed to identify treatment-induced changes in these sagittal plane motions. The averaged ankle and knee kinematic waveforms, consisting of 101 points were imported to the SPM1d software (version 0.4, available for download at http://www.spm1d.org/, accessed on 15 August 2022) in Matlab (The MathWorks, Natick, MA, USA, version 2020b). Clusters were formed if the critical threshold, based on random field theory, was crossed [50,82,83] (alpha = 0.05). In case of significant clusters, the p-value, details on the location and the duration (percentages of the entire gait cycle) were reported. Clusters were considered meaningful when their duration exceeded 3% of the gait cycle and when the differences between the means of the pre- and post-BoNT-A joint kinematics exceeded the respective SEMs [49,50].

## Figures and Tables

**Figure 1 toxins-14-00676-f001:**
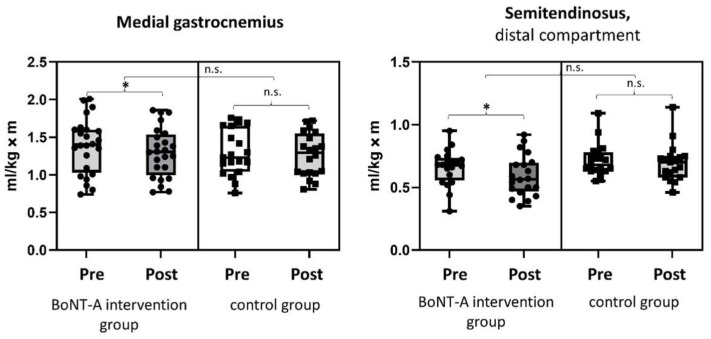
Box plots with individual data points of normalized muscle volume of the medial gastrocnemius and distal compartment of the semitendinosus before and after BoNT-A treatment in the BoNT-A intervention group and at the two assessments in the control untreated group. * alpha < 0.05, BoNT-A = Botulinum Neurotoxin-A, kg = kilograms, m = meters, ml = milliliters, n.s. = not significantly different.

**Figure 2 toxins-14-00676-f002:**
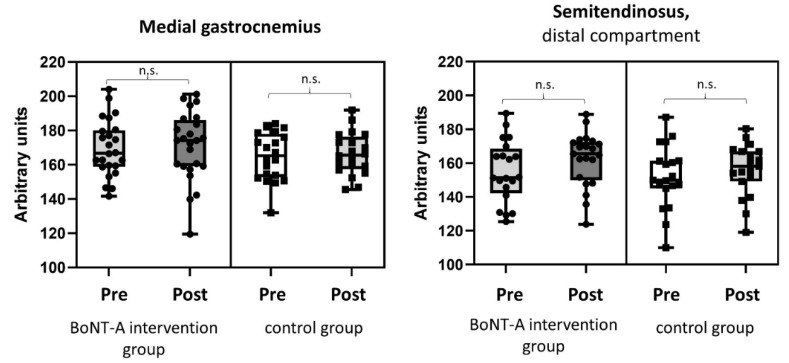
Box plots with individual data points of echogenicity intensity of the medial gastrocnemius and distal compartment of the semitendinosus before and after BoNT-A treatment in the BoNT-A intervention group and at the two assessments in the untreated control group. BoNT-A = Botulinum Neurotoxin-A, n.s. = not significantly different.

**Figure 3 toxins-14-00676-f003:**
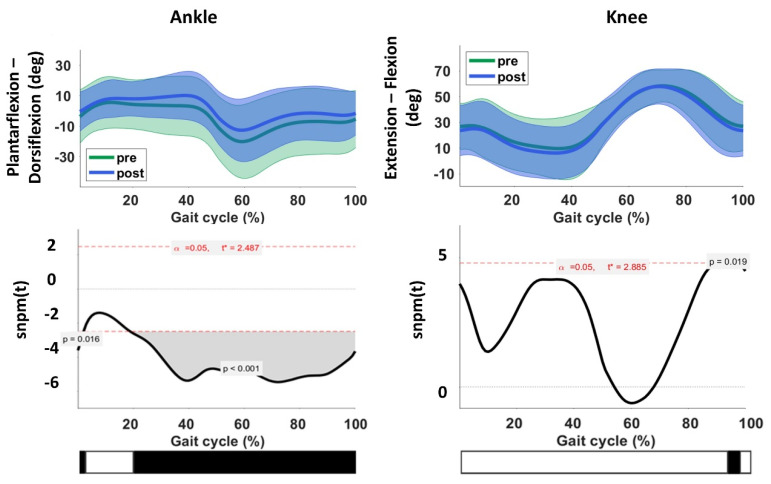
Overlay of the kinematic waveforms of the ankle and knee joint during gait, pre- and post-intervention. The green waveform represents the pre-treatment kinematics, and the blue waveform represents the post-treatment joint kinematics. The black bar at the bottom of the image indicates the location of the significant clusters, exceeding 3% of the gait cycle and exceeding the standard error of measurement. Deg = degrees. t = critical threshold, calculated by the random field theory.

**Figure 4 toxins-14-00676-f004:**
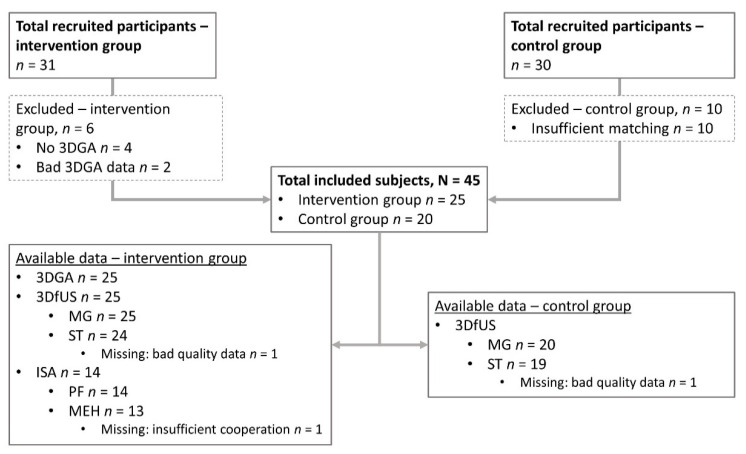
Flow chart of all recruited and finally included subjects in both the intervention and untreated control group, together with a summary of data availability. 3DfUS = three-dimensional freehand ultrasound, 3DGA = three-dimensional gait analysis, ISA = instrumented spasticity assessment, MEH = medial hamstrings, MG = medial gastrocnemius, *n* = number, PF = plantar flexors, ST = semitendinosus.

**Table 1 toxins-14-00676-t001:** Participant characteristics of the included participants of the intervention and control group.

	Intervention Group (*n* = 25)	Control Group (*n* = 20)	Between Group *p*-Value
Age (years) ^1^	6.4 (4.6–8.5)	7.6 (6.2–9.4)	0.144
Gender (*n*)	boys = 13, girls = 12	boys = 16, girls = 4	
GMFCS (*n*)	I = 14, II = 9, III = 2	I = 14, II = 5, III = 1	
Paralysis (*n*)	bilateral = 15 unilateral = 10	bilateral = 10 unilateral = 10	
Body height (cm) ^1^	114.5 (106.5–130.5)	127.3 (115.7–133.8)	0.021
Body mass (kg) ^1^	20.0 (17.0–23.9)	24.4 (21.1–30.7)	0.010
Maximal dorsiflexion angle, knee extended ^1^	5.0 (0.0–10.0)	10.0 (0.0–10.0)	0.267
Popliteal angle, unilateral ^1^	−55.0 (−42.5–−62.5)	−55.0 (−50.0–−63.8)	0.737
MAS PF, knee extended ^1^	2.0 (1.6–3.0)	1.5 (1.4–1.6)	
MAS PF, knee 90 degrees ^1^	2.8 (1.5–2.0)	1.5 (1.4–1.5)	
MAS hamstrings ^1^	1.5 (1.0–1.9)	1.0 (0.8–1.5)	
BoNT-A MG (units/kg), *n* = 24 ^1^	2.7 (2.2–2.9)	n.a.	
BoNT-A ST (units/kg), *n* = 21 ^1^	1.7 (1.4–2.0)	n.a.	
BoNT-A treatment history (*n*)	BoNT-A naive = 10 1–3 previous BoNT-A = 11 >4 previous BoNT-A = 4	BoNT-A naive = 9 1–3 previous BoNT-A = 10 >4 previous BoNT-A = 1	
Orthotic management, AFO’s during the day (*n*)	Frequently used (>50% of the day) = 20 Not frequently used (≤50% of the day) = 3 Not used = 2	Frequently used (>50% of the day) = 13 Not frequently used (≤50% of the day) = 2 Not used = 2 Unknown = 3	
Oral tone reduction (*n*)	Yes = 3 No = 22	Yes = 0 No = 20	

^1^ Values are presented as medians with 25th and 75th percentiles, AFO = ankle foot orthoses, BoNT-A = Botulinum Neurotoxin type-A, cm = centimeters, GMFCS = Gross motor function classification system, kg = kilograms, MAS = modified Ashworth score, MG = medial gastrocnemius, n = number, n.a. = not applicable, PF = plantar flexors, ST = semitendinosus.

**Table 2 toxins-14-00676-t002:** Muscle-specific growth rates calculated over the period between the two assessments for both the intervention and control group.

	Intervention Group	Control Group	Between-Group *p*-Value
Medial gastrocnemius growth rate (mL/month) ^1^	0.00 (−1.34–0.97)	0.17 (−0.23–0.87)	0.346
Semitendinosus, distal compartment growth rate (mL/month) ^1^	−0.49 (−1.45–0.58)	0.02 (−0.39–0.48)	0.226

^1^ Growth rates are presented as medians with corresponding 25th and 75th percentiles. mL = milliliters.

**Table 3 toxins-14-00676-t003:** Sagittal plane gait kinematic parameters, gait profile score and spatial-temporal parameters of the intervention group during the pre- and post-treatment assessments and difference scores.

Gait	Intervention Group			
Kinematic Parameters—Ankle ^1^	**Pre BoNT-A**	**Post BoNT-A**	**Difference Score**	***p*-value, treatment-induced/within group**
ankle ROM in sagittal plane (total GC) (degrees)	25.85 (19.83–34.55)	24.18 (21.51–29.13)	−1.74 (−9.04–3.07)	0.143
ankle angle in sagittal plane at IC (degrees)	−3.83 (−11.13–2.14)	−2.25 (−5.25–4.94)	3.56 (0.47–6.36)	0.002 **
maximum ankle angle (max DF) in sagittal plane during stance phase (degrees)	9.47 (0.70–14.52)	12.89 (7.37–17.76)	3.99 (−0.32–9.99)	0.004 **
maximum ankle angle (max DF) in sagittal plane during swing phase (degrees)	−1.71 (−9.63–3.12)	2.53 (−3.66–7.03)	5.34 (1.16–9.70)	<0.001 **
Kinematic parameters—Knee ^1^	**pre BoNT-A**	**post BoNT-A**	**difference score**	***p*-value, treatment-induced/within group**
knee ROM in sagittal plane (degrees)	53.50 (45.13–62.03)	54.84 (46.85–60.12)	1.94 (−3.79–8.84)	0.158
knee angle in sagittal plane at IC (degrees)	26.45 (20.93–35.43)	25.53 (17.26–30.53)	−1.69 (−9.19–1.84)	0.048
minimal knee angle in sagittal plane during stance (degrees)	5.06 (0.79–14.05)	6.74 (−1.22–10.09)	−2.44 (−7.61–2.23)	0.065
Gait profile score ^1^	**pre BoNT-A**	**post BoNT-A**	**difference score**	***p*-value, treatment-induced/within group**
Gait profile score for angles (degrees)	9.80 (7.41–11.33)	8.30 (6.71–9.41)	−1.76 (−2.51 – −0.15)	0.005 *
Spatial-temporal parameters ^1^	**pre BoNT-A**	**post BoNT-A**	**difference score**	***p*-value, treatment-induced/within group**
Cadence (number of steps/second)	2.25 (1.95–2.43)	2.19 (1.91–2.35)	−0.05 (−0.28–0.10)	0.156
Walking velocity (meter/second)	0.89 (0.67–1.10)	0.85 (0.73–1.05)	0.01 (−0.10–0.10)	0.911
Stride length (meters)	0.81 (0.66–0.96)	0.79 (0.69–1.04)	0.02 (−0.03–0.08)	0.108

^1^ Pre, post and difference scores are presented as median and 25th and 75th percentiles. A Bonferroni correction was applied for the kinematic parameters per joint (ankle: alpha 0.05/4 = 0.0125, knee: alpha 0.05/3 = 0.0167), * alpha < 0.05, ** significant different after Bonferroni correction: alpha < 0.0125 for the ankle and alpha < 0.0167 for the knee. DF = dorsiflexion, GC = gait cycle, IC = initial contact, max = maximum, ROM = range of motion.

**Table 4 toxins-14-00676-t004:** Outcomes of the instrumented spasticity assessment for a subgroup (*n* = 14) of the BoNT-A intervention group.

Instrumented Spasticity Assessment—Treated Muscles	Intervention Group (*n* = 14)			
Parameter	pre BoNT-A	post BoNT-A	Difference Post-Pre	*p*-Value, Treatment-Induced/within Group
Medial gastrocnemius, EMG—absolute high RMS–EMG (µV)	11.52 (4.87–22.07)	4.91 (3.67–9.45)	−4.73 (−15.74–1.31)	0.019 *
Medial hamstrings, EMG—absolute high RMS–EMG (µV)	12.07 (5.77–16.04)	7.98 (3.68–10.58)	−2.82 (7.64–0.21)	0.041 *

* alpha < 0.05, EMG = electromyography, RMS = root mean square, µV = microvolts.

## Data Availability

The datasets generated for this study are available on request to the corresponding author. The raw data supporting the conclusions of this manuscript will be made available by the authors, without undue reservation, to any qualified researcher.

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
