# Peer review of "The Short-Term Impact of Botulinum Neurotoxin-A on Muscle Morphology and Gait in Children with Spastic Cerebral Palsy"

_toxins, 2022, doi:10.3390/toxins14100676_

Round 1

Reviewer 1 Report

My comments and suggestions can be found in the attached file.

Author Response

We would like to thank the reviewer for providing us with the feedback. We answered his/her questions to the best of our ability and made changes in the manuscript if this was warranted.

A detailed response to the reviewer’s questions can be found in the attached file. 

The reviewer’s questions are indicated in regular font and our answers in italic. Any citations of the revised manuscript are listed between quotation marks. All revisions to the manuscript are marked up using the “Track Changes” function.

Reviewer 2 Report

In the present study, the authors investigated the effect of BoNT-A on muscle morphology, spasticity and gait using an integrated set of objective assessments. They hypothesized that BoNT-A injections would result in reduced spasticity and improved gait, but would induce muscle atrophy.

Unfortunately, this manuscript needs very substantial improvements and corrections before publishing may be possible.

General points:

Please add a list of abbreviations before References section to your manuscript.

Please do your List of references according to “Toxins”.

Special points:

 Keywords: please add also to keywords: treatment

 Introduction

Lines 28-40: please add multiple references at the end of each these sentences.  

Lines 41-50: please describe exactly all these studies.

Lines 66-78: please describe exactly all these studies.

 Main part of the manuscript:

Materials and Methods

Lines 503-509: please add the exact organisation name, date and number of the permission for all your experiments.

Please describe exactly treatment with BoNT-A and exactly injection points and add the appropriate literature.

Please add to each Figure the Legend with description.

Please add to your manuscript the Future perspectives section.  

Author Response

(The authors gave the same response as above.)

Round 2

Reviewer 1 Report

I thank the authors for their significant work in responding to my comments and questions and in modifying the manuscript.

Author Response

We thank the reviewer for his/her positive feedback and for accepting our modifications to the manuscript.

Reviewer 2 Report

Thank you for your corrections. Once again, please add to each Figure a Legend with exactly description of this Figure and your results.

Please correct your List of References according to "Toxins".

Author Response

Our comments to the reviewer's suggestions can be found in the attached file.
